

# Intercomparison of TCCON data from two Fourier transform spectrometers at Lauder, New Zealand.

David F. Pollard, John Robinson, Hisako Shiona, and Dan Smale

National Institute of Water & Atmospheric Research Ltd (NIWA), Lauder, New Zealand

**Correspondence:** David F. Pollard (dave.pollard@niwa.co.nz)

**Abstract.** We describe the change of operational instrument for the routine measurement of column-averaged dry-air mole fraction of several greenhouse gases (denoted $X_{gas}$) at the Lauder Total Carbon Column Observing Network (TCCON) site and the steps taken to demonstrate comparability between the two observation systems following a systematic methodology.

Further, we intercompare retrieved $X_{gas}$ values during an intensive intercomparison period during October and November
2018, when both instruments were performing optimally, and on subsequent, less frequent occasions. The average difference between the two observing systems was found to be well below the expected level of uncertainty for TCCON retrievals for all compared species. In the case of $X_{CO_2}$ the average difference was $0.0264 \pm 0.0465\%$ ($0.11 \pm 0.19\,\mu mol\,mol^{-1}$).

## 1 Introduction

The Total Carbon Column Observing Network (TCCON, Wunch et al. (2011)) coordinates globally distributed measurements
of near infrared solar absorption spectra from which high precision retrievals of the column-averaged dry-air mole fraction of several greenhouse gases (denoted $X_{GAS}$), including $CO_2$, $CH_4$ and $CO$, can be made.

The National Institute of Water & Atmospheric Research (NIWA) atmospheric observatory at Lauder, New Zealand, was one of the first TCCON sites and has been operating since 2004. The site initially used a Bruker IFS 120HR (serial number 39, TCCON identifier lh) Fourier transform spectrometer (FTS) to take both near infrared (NIR) TCCON measurements and mid
infrared (MIR) observations for the Network for the Detection of Atmospheric Composition Change Infrared Working Group (NDACC-IRWG, De Mazière et al. (2018)). This meant that there were regular instrument interventions to change optical components. In 2010 a dedicated Bruker IFS 125HR (serial number 072, TCCON identifier ll) was purchased to continue the TCCON measurements in parallel with MIR measurements on the IFS 120HR. The history of the instrument systems used for the TCCON dataset, as well as a thorough description of the site, retrieval scheme and validation of the dataset were previously
presented in Pollard et al. (2017), hereafter referred to as Pollard17, and a summary of the instrument changes is given in Table 1.

Because the Bruker IFS 120HR became unsupported by the manufacturer, it was decided to purchase a second IFS 125HR (serial number 132, lr) to continue the TCCON dataset and switch the existing instrument to MIR measurements for the NDACC to ensure the continued reliability of both data sets.



The purpose of this article is to define the testing and comparisons that needed to be undertaken in order to ensure that the two instrument systems give comparable results and to demonstrate that the Lauder TCCON dataset meet these requirements and can be considered continuous across the change of instruments.

    There have been several past studies which have compared the measurements of low resolution, portable Bruker EM27/SUN FTS instruments with the IFS 125HRs of TCCON stations, e.g. Gisi et al. (2012) and Hedelius et al. (2016). Side-by-side

comparisons of high resolution instruments are less common. Batchelor et al. (2009) intercompared MIR measurements from a Bruker IFS 125HR with a Bomem DA8 at the NDACC-IRWG site at Eureka, Canada, Messerschmidt et al. (2010) were able to compare NIR measurements from two IFS 125HR instruments side-by-side at the TCCON site in Bremen, Germany and the comparison of the IFS 120HR and the original IFS 125HR at Lauder was described in Pollard17.

    The work described here represents the first time that an operational TCCON station has changed measurements between

two IFS 125HR instruments and describes the steps needed to ensure comparability of their measurements.

    In the next section we will briefly describe the instrumentation and retrieval schemes. Section 3 will outline the tests undertaken to ensure comparability between the retrievals carried out using both instruments. Conclusions will be drawn in Sect. 4.

## 2 Experimental setup

In this section we outline both the instrumentation and the retrieval scheme used to produce the Lauder TCCON site dataset. This has already been described in detail in Pollard17, therefore this section will give a broad overview and concentrate on details specific to the change of instrument.

### 2.1 Instrumentation and data collection

The Bruker Optik GmbH IFS 125HR FTS is the primary instrument of the TCCON. Over the course of the Lauder TCCON

site time series we have measured using three instruments as outlined in the introduction and detailed in Table 1. For clarity hereafter we will refer to the instruments by their two letter TCCON site identifier (i.e. lr for the new 125HR, ll for the previous 125HR and lh for the original 120HR which will not be discussed in detail herein).

    The two instruments compared in this work are functionally identical, using a calcium fluoride beam-splitter, a 45 cm path difference to give a spectral resolution of $0.02\,\mathrm{cm^{-1}}$. The DC output of two detectors, InGaAs (spectral range $3800 - 12000\,\mathrm{cm^{-1}}$)

and Silicon ($9000 - 16000\,\mathrm{cm^{-1}}$), are measured simultaneously.

    The high-resolution FTS instruments at Lauder are accommodated in a purpose built, temperature-controlled building. In May 2018 instrument lh was removed from the building and replaced by lr, leaving ll in its original position.

    Each instrument is positioned below a dedicated solar tracker with optical feedback providing a pointing accuracy of $0.02°$ (Robinson et al., 2020).

Ancillary meteorological measurements are made at a nearby climate station and the pressure data from this are necessary for the GHG retrievals.





Through the use of automatic scheduling software (Geddes et al., 2018), the continuous operation of the solar trackers and the use of automated tracker covers which close a hatch over the solar pointing elevation mirror in the presence of precipitation or winds above a certain threshold, the operational TCCON instrument (lr) is able to make unattended measurements at any time.

During the intensive intercomparison period between October and November 2018, the ll instrument was also left configured for NIR measurements and able to operate unattended in parallel with lr. Since November 2018 intercomparison measurements have been conducted on ll on an opportunistic basis. This has resulted in 34 days where both instruments were recording NIR spectra, spread across 12 months to September 2019.

## 2.2   Retrieval scheme

The GGG suite of processing software, currently version GGG2014 as described by Wunch et al. (2015), is used across the TCCON and includes software to process raw interferograms to spectra (i2s) and a non-linear, least squares fitting algorithm (GFIT). The implementation of GGG2014 used for the lr instrument is the same as for ll and has previously been described in Pollard17.

It is important to note that the resulting outputs of the retrieval scheme are dry air mole fractions (DMFs or $X_{gas}$), where

the vertical column of the retrieved gas is scaled by the co-retrieved vertical column of oxygen in order to remove instrumental biases:

$$X_{gas} = \frac{VC_{gas}}{VC_{O_2}} \times 0.2095 \qquad (1)$$

Where 0.2095 is the assumed dry-air mole fraction of $O_2$. The DMF of dry-air, $X_{air}$ is a special case given by:

$$X_{air} = \frac{VC_{air}}{VC_{O_2}} \times 0.2095 - X_{H_2O} \times \frac{m_{H_2O}}{m_{air}^{dry}} \qquad (2)$$

where $m_{H_2O}$ and $m_{air}^{dry}$ are the mean molecular masses of water (18.02 g mol[-1]) and dry-air (28.964 g mol[-1]), and $VC_{air}$ is calculated from the surface pressure, $P_s$:

$$VC_{air} = \frac{P_s}{\{g\} \times \frac{m_{air}^{dry}}{N_a}} \qquad (3)$$

Where $\{g\}$ is the column-averaged acceleration due to gravity and $N_a$ is Avogadro's constant.

In an idealised case $X_{air}$ would be unity, but limitations in the spectroscopic databases used for the retrievals mean that

the actual value typically lies within 1% of 0.98. The value and stability of $X_{air}$ is used as a diagnostic of the measurement system as it is not scaled by the $O_2$ column, therefore deviations from the nominal value can be indicative of instrumental and systematic problems such as timing or pointing errors.



## 3    Comparison tests and results

Hedelius et al. (2016) attempted to identify all parts of the measurement and retrieval system that could lead to differences in
the retrieved $X_{gas}$ quantities of two different FTS systems, which they summarised in Table 6. of that article and we have used
this as the basis for systematically demonstrating the comparability of the two instrument systems.

Several factors listed in Table 6 of Hedelius et al. (2016) are not relevant to the intercomparison being considered in this
work for following the reasons:

- Because the two instruments are functionally identical, the incoming radiation attenuation effect, optimum averaging
  time and resolution effects do not need to be considered.

- Solar zenith angle (SZA) artefacts are negated by comparing temporally coincident observations made in parallel.

- Spectral fitting windows and the uncertainty budget for the fitting algorithm do not need to be considered because the
  same retrieval scheme is used for both instruments. The same is also true for the averaging kernels. However, these will
  have a dependence on the instrument signal-to-noise ratio (SNR) but this will be a lower order effect than the variation
  with SZA (Wunch et al., 2011), especially as the SNR is very similar for both instruments (see Sect. 3.1). Therefore it
  need not be considered in this work.

- Long-term artefacts are not relevant over the period of this study.

- Because the instruments are co-located, region/zone dependence, surface pressure effects, sensitivity to the profile of
  meteorological parameters and differences in the *a priori* can be ignored.

In the subsections below, we first examine the signal-to-noise characteristics of the two instruments and then address each of
the remaining items in Table 6 of Hedelius et al. (2016) in its own subsection.

### 3.1    Signal-to-noise ratio

There are several of methods for calculating the signal-to-noise ratio (SNR) of a spectrum. In this section we use the method
implemented in the upcoming version of the GGG processing suite, GGG2020, which smooths the spectrum to remove instru-
mental noise in order to calculate the signal level and then compares the RMS differences of the unsmoothed spectrum with
the smoothed spectrum in regions where the signal is close to zero.

Figure 1 shows histograms of the spectral SNRs calculated for both instruments during October 2018. Only spectra which
cleared the initial GGG quality checks (convergent solution and volume mixing ratio scaling factor, RMS fit residuals, fre-
quency shift and solar gas shift within thresholds) were included in the statistics and outliers are not shown. The median SNR
for ll is 154 and for lr, 157 and the means are 153.5 and 154.1 respectively (standard deviations 8.2 and 10.3 respectively). The
lr SNRs have a larger number of low value outliers resulting in the lower value for the mean. However, the median values are
similar and we conclude that the two instruments perform to a similar standard in this regard.



## 3.2 Instrument lineshape

The instrument lineshape (ILS) retrieved from lamp measurements of a gas cell containing a known amount of HCl is used as a
diagnostic of the alignment and stability of instruments across the TCCON. This is achieved using the LINEFIT 14.5 software
and methodology outlined in Hase et al. (2013).

Since Pollard17, the retrieval settings used at Lauder to obtain the ILS have changed from one which described the ILS in
terms of two typical misalignment parameters (shear and angular) to one which fully retrieves the ILS as a function of optical
path difference (OPD), in accordance with TCCON-wide guidance.

Over the period presented here, the mean modulation efficiency (ME) at the maximum OPD for ll is $1.022 \pm 0.002$ and the
maximum phase error (PE) is $0.002 \pm 0.002\,rad$. This represents an increase in ME at max. OPD and a reduction in max. PE
to the values presented in Pollard17, following a realignment of ll in October 2017. The 2.2% overmodulation at max. OPD
however, remains below the 4% required to ensure the necessary retrieval accuracy for $X_{CO_2}$ (Hase et al., 2013). For lr, the
mean ME at max. OPD is $0.994 \pm 0.005$ and the mean max. PE is $-0.002 \pm 0.001\,rad$.

Figure 2 shows both the ME at the maximum OPD and the maximum PE for both instruments at approximately monthly
intervals during and since the intercomparison period. This demonstrates the quality and stability of the alignment of both
instruments.

## 3.3 Laser sampling error

It is a known feature of the IFS 125HR instruments that the metrology laser can be sampled incorrectly resulting in some of the
spectral information above the Nyquist frequency of $7899\,cm^{-1}$ being folded below it and vice versa, causing features known
as "ghosts" (Messerschmidt et al., 2010). This is mitigated in two ways. Firstly, the zero level on the laser amplifier board is
tuned to minimise the effect and subsequently checked annually, a process more fully described in Pollard17. Secondly, within
the GGG2014 i2s software, the spectra are re-sampled based on the spectra of the silicon detector, which is wholly contained
in the upper half of the alias as described in Wunch et al. (2015).

Figure 3 shows the laser sampling error (LSE) determined using this method for both instruments during the period of
the intercomparison, showing a mean and standard deviation of $1.475 \pm 1.315 \times 10^{-4}$ and $1.167 \pm 1.612 \times 10^{-4}$ for ll and lr
respectively. These diagnosed values are small relative to the range of LSE that can be resampled by i2s, and therefore will not
have a detrimental effect on retrievals.

## 3.4 Frequency shifts

The absolute calibration of the measured spectral grid can be affected by either a discrepancy between the actual and expected
laser wavenumber of the metrology laser or a Doppler shift of the absorbing species in the atmosphere caused by atmospheric
motion parallel to the solar pointing direction.

GGG retrieves this frequency shift from the idealised spectroscopy of the telluric absorption features for each micro-window.
For the purposes of this comparison we choose to examine the fitted shift in the oxygen window centred at $7885\,cm^{-1}$ as this is



the broadest micro-window and thus limits the sensitivity to specific species. For the one-month period of the intercomparison we find that the median shift for the ll instrument is $-0.469\,\Delta\nu/\nu\times10^6$ (standard deviation 0.028) and for lr is $-0.507\,\Delta\nu/\nu\times10^6$ (0.026). This demonstrates similar performance of the metrology laser in both instruments. The variability of the shift is likely dominated by the wind induced Doppler shift, hence the similarity in the standard deviation values.

### 3.5   Solar gas shifts

Similarly, to the frequency shift, GFIT retrieves the shift of the solar spectroscopic lines from their expected value. GFIT accounts for the Doppler induced shift caused by the Earth's rotation and orbital eccentricity and so the remaining shift is wholly due to the Doppler shift induced by the Sun's rotation if the instrument solar tracker is not pointed at the centre of the solar disc.

    Figure 4 shows the retrieved solar gas shift (SGS) as a function of solar zenith angles for both instruments on 7th February
2019. Also plotted is the equivalent pointing accuracy required to achieve the TCCON target precision. As can be seen, the retrieved SGS remains well within this limit throughout the day, despite a small deviation for lr at high solar zenith angles in the morning as the solar tracker achieved a lock on the Sun and transitioned from passive to active tracking, indicating acceptable solar pointing is achieved.

    This provides confidence that the performance of the entire measurement system: in this case the solar tracker, FTS and
retrieval scheme, are similar for both instruments. A more thorough discussion of the solar tracking system and its assessment is provided in Robinson et al. (2020).

### 3.6   Airmass dependence

Due to spectroscopic limitations, the retrievals of $X_{CO_2}$ and a number of other species exhibit a SZA or airmass dependence at all TCCON sites. An airmass dependent correction factor (ADCF) is derived for these species following Appendix A(e)
of Wunch et al. (2011) and is based upon fitting a symmetric and anti-symmetric function to the diurnal variation about the mean value. It is assumed that the symmetric variation is likely to be an artefact due to limitations of the spectroscopy used in the retrieval and the anti-symmetric component is real. For the TCCON-wide correction an ADCF is computed based upon long-term retrievals from a subset of sites.

    The ADCFs computed for both instruments during the October 2018 comparison period, and the TCCON-wide values, are
shown in Table 2. Because the symmetric term can also be affected by instrumental problems (e.g. zero level offsets, continuum curvature and ILS uncertainties) it is reassuring that the ADCFs derived for both instruments are consistent with one another and the prescribed TCCON values.

### 3.7   $X_{gas}$ comparison

In this section we present data from both instruments retrieved from measurements made during the October - November 2018
intercomparison period and intermittently since.





In order to make meaningful comparisons between the two instruments, the data are first averaged over ten-minute bins. Ten minutes was chosen in order to ensure that temporally coincident values are being compared whilst not aliasing in effects due to airmass dependence or natural variability.

This method results in 833 ten-minute averages being compared from both instruments. Correlation plots for $X_{AIR}$, $X_{CO_2}$,

$X_{CH_4}$ and $X_{CO}$ are shown in Figure 5 and summarised in Table 3.

For $X_{air}$ lr is on average 0.0855% higher than ll (standard deviation 0.1272), and the spread of the difference increases slightly at higher SZAs as shown in Figure 6. This is likely caused by small differences in the time it takes both instruments to conduct a measurement, due to slightly different firmware versions or hardware, leading to a small difference in the computed airmass for the forward and reverse scans. This is an effect which is amplified at high SZAs when the airmass is changing more

rapidly, as detailed in Pollard17.

The average $X_{CO_2}$ is virtually the same for both instruments (ll is 0.0264% higher than lr with a standard deviation of the differences of 0.0465%) and well within the expected uncertainty of the retrieval scheme of 0.25% and the target site-to-site bias of 0.2%. As can be seen in Figure 7, there is no discernible variation with SZA, as the small timing error effect will have been negated during the scaling by the vertical column of $O_2$ to derive the dry air mole fraction.

## 4 Conclusions

We have taken a systematic approach to demonstrating the comparability of retrieved quantities of $X_{CO_2}$, $X_{CH_4}$ and $X_{CO}$ from an existing and new Bruker IFS 125HR instruments at the Lauder TCCON site. The approach adopted considered each instrument system, including the solar tracker and the processing and retrieval scheme, as a whole.

Most potential sources of discrepancy, both instrumental and methodological, can be discounted due to the co-location of

the two instruments and the use of a common processing and retrieval scheme. For the remaining, instrument specific, sources we addressed each one to assure comparability.

Finally, we compared the retrieved data from each instrument over a one month comparison period in October 2018 and find excellent agreement with the average difference between 833 ten-minute averages of 0.0264% for $X_{CO_2}$ (ll-lr, standard deviation 0.0465%), which is well below the expected TCCON site-to-site bias of 0.2%. The difference in $X_{CO_2}$ reported

here also compares favourably with previous work. In Pollard17, the comparison of the lh and ll instruments showed a mean difference in daily averages of $X_{CO_2}$ of $0.068 \pm 0.113\%$ and Messerschmidt et al. (2010) reported an average difference for one hour of data of $0.07\%$.

We therefore conclude that users of the Lauder TCCON dataset can consider it to be continuous across the change of instruments.

*Code and data availability.* The Lauder TCCON data can be downloaded from the TCCON Data Archive (https://tccondata.org/) and can be individually cited as Sherlock et al. (2014b) and Pollard et al. (2019), the data available on the archive includes retrievals from both ll





and lr for the month of October 2018. Further ll intercomparison data beyond this period are available from the authors. The GGG software package can be downloaded from the TCCON wiki pages (https://tccon-wiki.caltech.edu/). LINEFIT can be obtained from the Karlsruhe Institute of Technology: https://www.imk-asf.kit.edu/english/897.php.

*Author contributions.* JR was responsible for the specification, procurement, installation and testing of both instruments compared herein. DS contributed to the design and implementation of the testing and comparison methodology. HS develops and maintains the routine processing and retrieval software. DP performed the analysis and wrote this manuscript. JR, DS and DP collected the data. All authors have read and provided feedback on the paper.

*Competing interests.* The authors declare that they have no conflict of interest.

*Acknowledgements.* The authors would like to thank Gregor Surawicz and David Marston of Bruker Optik GmbH for their assistance with the procurement and commissioning of the new instrument. The Lauder TCCON programme is funded by NIWA through New Zealand's Ministry of Business, Innovation and Employment's Strategic Science Investment Fund with additional support from the National Institute for Environmental Studies, Japan, GOSAT project.





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





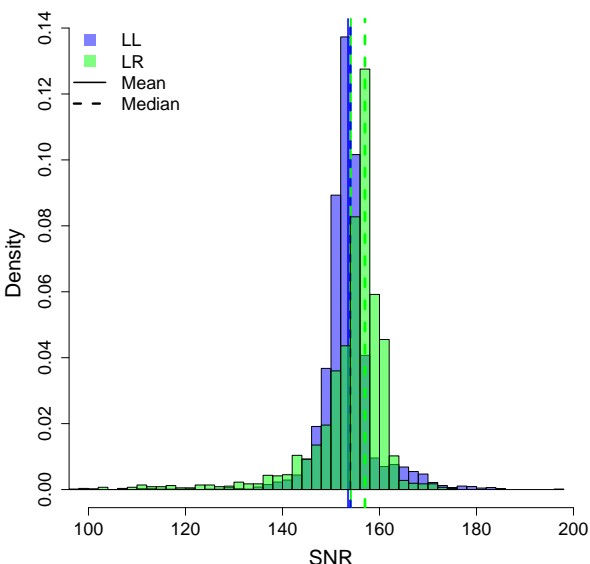

**Figure 1.** Histogram of spectral SNR for both instruments during October 2018

**Table 1.** High resolution Fourier Transform spectrometers used at Lauder

| Instrument model | Bruker SN | NDACC ID | TCCON ID (data reference) | Previous role | Current role |
|---|---|---|---|---|---|
| IFS 120HR | 39 | NIWA001 | lh (Sherlock et al., 2014a) | NDACC (2004-2018) | Research (2018+) |
| IFS 125HR | 72 | NIWA006 | ll (Sherlock et al., 2014b) | TCCON (2010-2018) | NDACC (2018+) |
| IFS 125HR | 132 | NIWA008 | lr (Pollard et al., 2019) | - | TCCON (2018+) |

Sherlock, V., Connor, B., Robinson, J., Shiona, H., Smale, D., and Pollard, D. F.: TCCON data from Lauder (NZ), 125HR, Release GGG2014.R0, https://doi.org/10.14291/TCCON.GGG2014.LAUDER02.R0/1149298, https://data.caltech.edu/records/281, 2014b.

Wunch, D., Toon, G. C., Blavier, J. F. L., Washenfelder, R. A., Notholt, J., Connor, B. J., Griffith, D. W. T., Sherlock, V., and Wennberg, P. O.: The Total Carbon Column Observing Network, Philosophical Transactions of the Royal Society a-Mathematical Physical and Engineering Sciences, 369, 2087–2112, https://doi.org/10.1098/rsta.2010.0240, http://rsta.royalsocietypublishing.org/content/roypta/369/1943/2087.full.pdf, 2011.

Wunch, D., Toon, G., Sherlock, V., Deutscher, N. M., Liu, C., Feist, D. G., and Wennberg, P. O.: The Total Carbon Column Observing Network's GGG2014 Data Version, https://doi.org/doi:10.14291/tccon.ggg2014.documentation.R0/1221662, 2015.






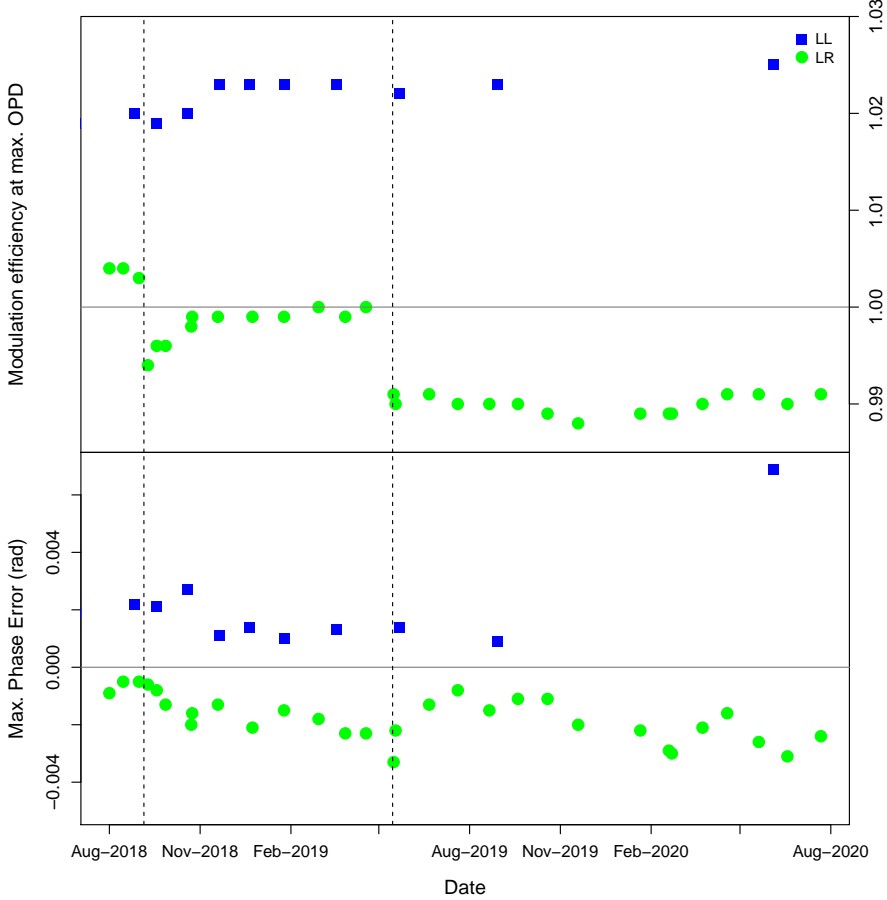

**Figure 2.** Time series of ILS retrievals for both instruments during the extended intercomparison period. Modulation efficiency at the maximum optical path difference (top panel) and the maximum phase error (lower panel). The dashed vertical lines indicate the following technical interventions of LR: 18[th] September 2018, the final alignment; and 28[th] May 2019, replacement of the metrology laser. There were no significant interventions to LL during this period.

**Table 2.** Comparison of derived airmass correction factors and their standard deviations for each instrument and the TCCON wide values

| Species | ll | | lr | | TCCON | |
|---|---|---|---|---|---|---|
| | ADCF | sd | ADCF | sd | ADCF | sd |
| $X_{CO_2}$ | -0.0087 | 0.0014 | -0.0093 | 0.0019 | -0.0068 | 0.0050 |
| $X_{CH_4}$ | -0.0006 | 0.0050 | 0.0008 | 0.0041 | 0.0053 | 0.0080 |
| $X_{N_2O}$ | -0.0023 | 0.0088 | -0.0002 | 0.0071 | 0.0039 | 0.0100 |
| $X_{CO}$ | -0.0712 | 0.0619 | -0.0538 | 0.0535 | -0.0483 | 0.1000 |

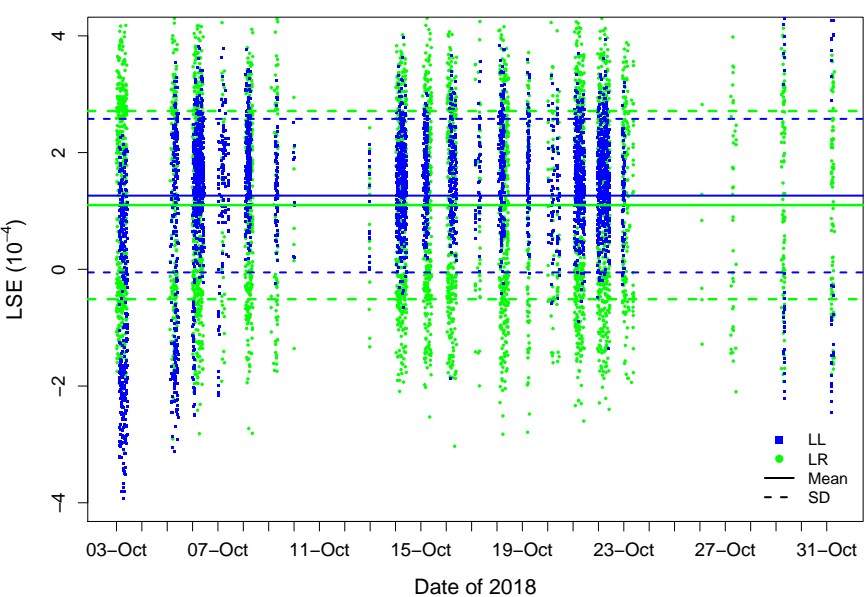

**Figure 3.** Laser sampling error (LSE) diagnosed by i2s for both instruments during October 2018.

**Table 3.** Results from the comparison of 833 ten-minute averages (with 5 or more measurements per instrument) for individual species between ll and lr.

| Species | Mean difference (ll-lr) % | Standard deviation |
|---|---|---|
| $X_{AIR}$ | -0.0855 | 0.1272 |
| $X_{CO_2}$[*] | 0.0264 | 0.0465 |
| $X_{CH_4}$ | -0.0561 | 0.0647 |
| $X_{CO}$ | 0.0852 | 0.5264 |

[*]TCCON site-to-site bias target = 0.2%

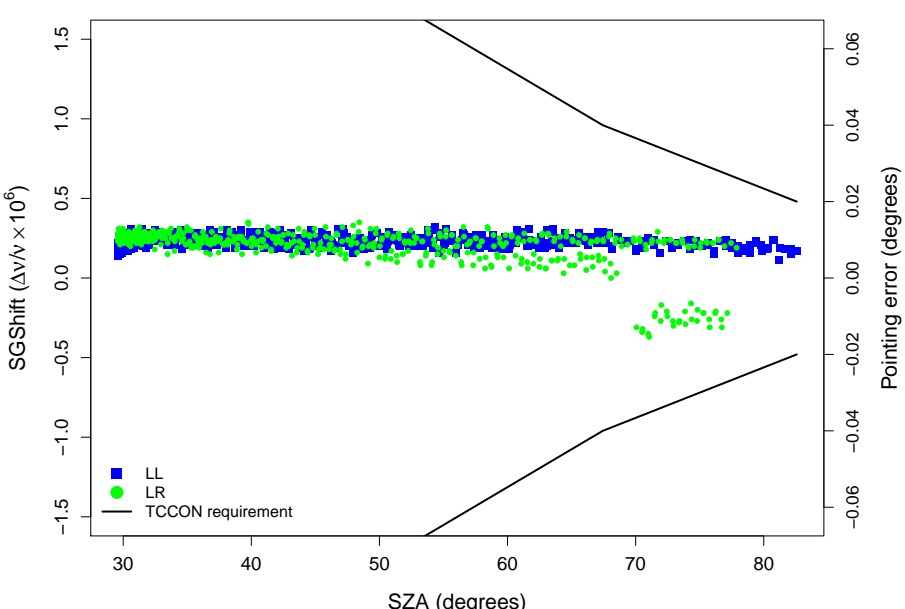

**Figure 4.** Retrieved solar gas shift (left axis) and corresponding angular pointing error (right axis, assuming that any mispointing is perpendicular to the Sun's axis of rotation) as a function of solar zenith angle for both instruments during the course of 7[th] February 2019. The pointing accuracy required to maintain the TCCON precision target equivalent to a 0.2% error in $X_{CO_2}$ is indicated by the black line.



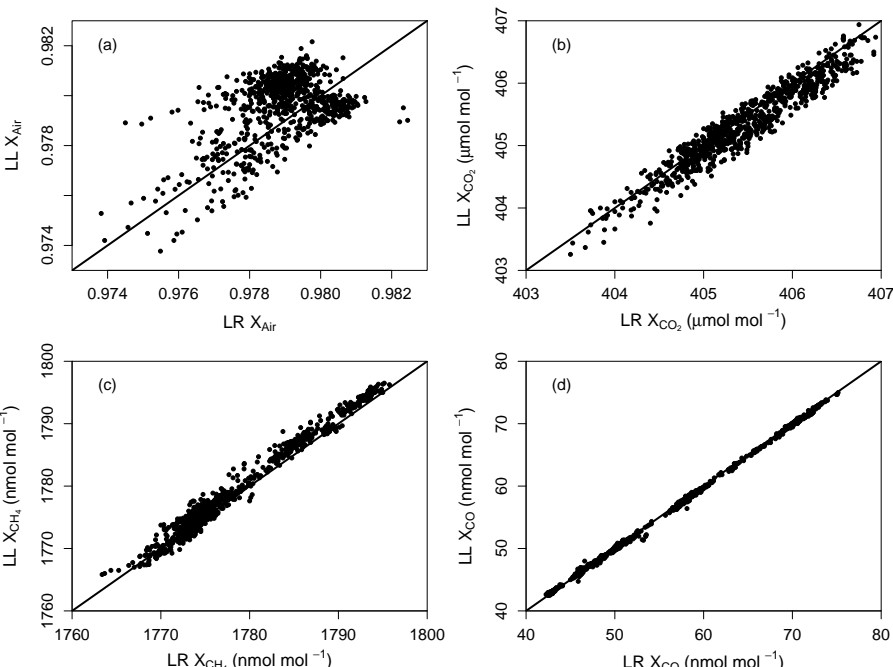

**Figure 5.** Correlation plots for 833 ten-minute averages of (a) $X_{AIR}$, (b) $X_{CO_2}$, (c) $X_{CH_4}$ and (d) $X_{CO}$. The solid black line represents the 1:1 relationship.





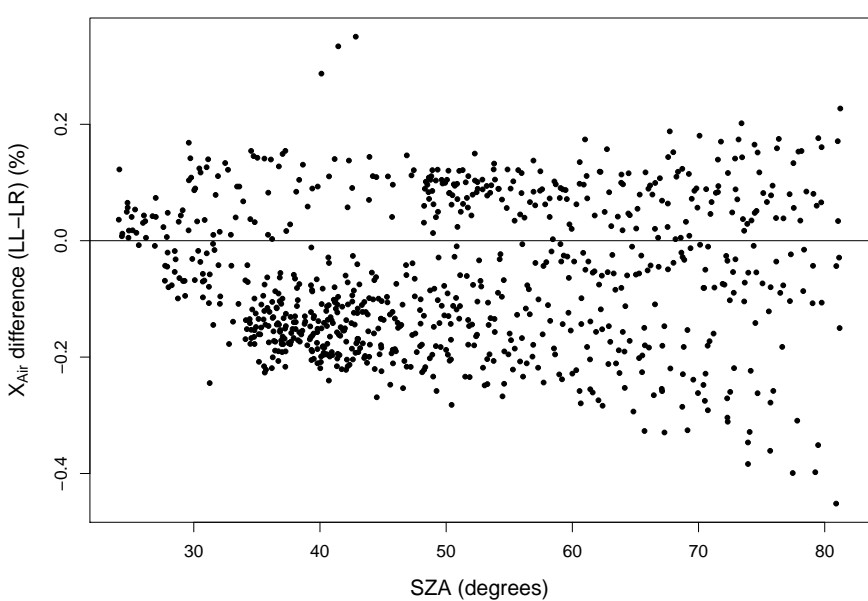

**Figure 6.** Difference (LL-LR) of ten-minute averages of retrieved $X_{air}$ expressed as a percentage, as a function of SZA.

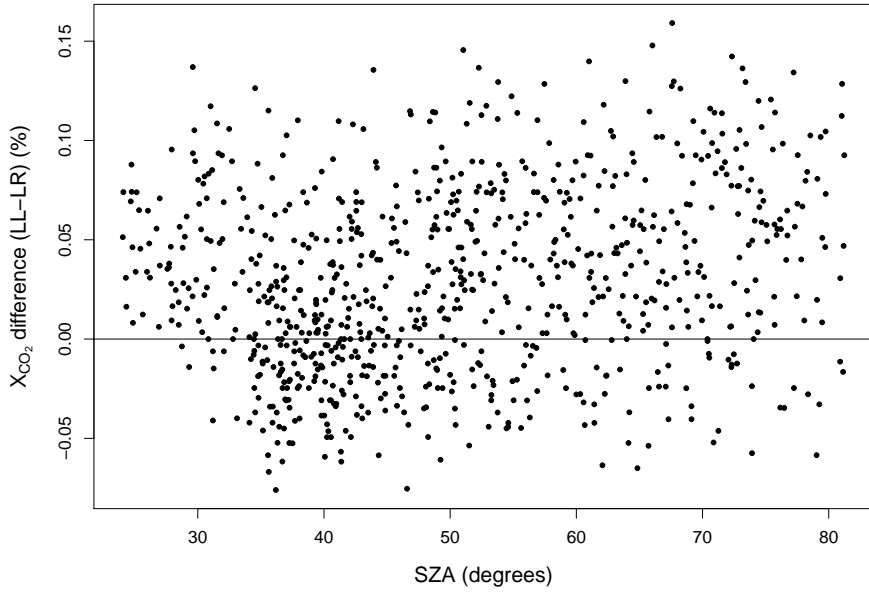

**Figure 7.** Difference (LL-LR) of ten-minute averages of retrieved $X_{CO_2}$ expressed as a percentage, as a function of SZA.