# Peer review of "Intercomparison of TCCON data from two Fourier transform spectrometers at Lauder, New Zealand."

_Atmospheric Measurement Techniques, 2020_

## Referee Comment (RC1) · Anonymous Referee #1 · 11 Nov 2020

The paper "Intercomparison of TCCON data from two Fourier transform spectrometers at Lauder, New Zealand" by Pollard et al. presents an intercomparison of two high-resolution Fourier transform spectrometer measurements to assure the continuity of the Lauder TCCON data. Pollard et al. demonstrate that the difference between the column-averaged dry-air mole fraction of carbon dioxide ($X_{CO_2}$) data obtained from the two instruments is well below the uncertainty of the TCCON product.

The Lauder TCCON data have been widely used for carbon cycle studies and validation of satellite-based greenhouse gas and carbon monoxide measurements. The topic of this paper is significant for those research fields and well suited to *Atmospheric Measurement Techniques*. This paper is concisely written and contains a full description of the instrumental intercomparison. I therefore recommend publication of this paper after correcting and addressing several minor concerns below.
* * *
Specific comments

L80-81: $X_{air}$ is scaled by the $O_2$ column because Equation (2) can be rewritten as follows:
$$X_{air} = \left( VC_{air} - VC_{H_2O} \frac{m_{H_2O}}{m_{air}} \right) \frac{0.2095}{VC_{O_2}}$$

The reason $X_{air}$ is used as a diagnostic of the measurement system is that the ratio between the retrieved columns is not taken for $X_{air}$.

L146: The median shift relative to the central wavenumber $\Delta\nu/\nu$ is $-0.469 \times 10^6$ ($-0.469 \Delta\nu/\nu \times 10^6$ is not the median shift). In addition, please define the variables $\Delta\nu$ and $\nu$ or $\Delta\nu/\nu$.

L154: Please clarify what the solar gas shift (SGS) means, in relation to just above sentence [GFIT accounts for …].

L184: It is unclear why "a small difference in the computed airmass for forward and reverse scans" induces the difference between the $X_{gas}$ data from the two instruments. Do the authors mean "a small error in the computed airmass (i.e., an error in zero path difference time)"?

L186-187: Please cite references for the values of the expected uncertainty of the retrieval scheme (0.25%) and the target site-to-site bias (0.2%). Provided that there are expected uncertainties of the retrieval scheme and target site-to-site bias for $X_{CH_4}$ and $X_{CO}$, I recommend specifying a similar evaluation here.

L197: October 2018 -> October and November 2018 (to be consistent with Abstract and Introduction)

Caption of Table 1: Transform -> transform

---

## Referee Comment (RC2) · Anonymous Referee #2 · 19 Nov 2020

Review of Pollard et al "Intercomparison of TCCON data from two Fourier transform spectrometers at Lauder, New Zealand" for AMT.

The paper by Pollard et al describes the intercomparison of two collocated Bruker FTIR high resolution spectrometers. The instruments are operated within the Total Column Carbon Observing Network (TCCON). This network has well controlled analysed procedures (GFIT suite of software), as well as agreed upon instrumentation (Bruker 125HR), and measurement protocols. The NZ team is very experienced in both measurements and analysis procedures demanded by TCCON. They are actively involved in the TCCON network in terms of running their own site and contributing to the success

of this network. On this basis this team is well placed to compare these instruments, one a new introduced FTIR, comparing the new one with an older established dataset. Their attention to detail is very good.

The text is well written, and as far as this referee can find, only one misplaced word (remove the first "of" in line 102). The authors establish that the measurement conditions are such that the comparison of the two datasets is relatively straightforward, that is, the conditions under which the data is collected is very similar in terms of instruments, collocation, and hence atmospheric conditions. They systematically consider the important nuances that have been carefully scrutinised and worked through over the years within the TCCON community, including Ghosts, airmass dependence, frequency shifts, signal to noise etc. The paper demonstrates that under normal conditions experienced at Lauder these two instruments perform at a remarkably consistent level, more than meeting various TCCON metrics.

The only suggestion here is a straightforward statistical one. Since the main product that is compared, the means of the various retrieved Xgas, a simple t-test would give a solid quantitative basis to the conclusion that both instruments are measuring the same thing.

This paper is recommended for publication in AMT.

---

## Author Comment (AC1) · 12 Jan 2021

We thank the reviewer for taking the time to review our manuscript and for their constructive and thought provoking comments.

Below we have included the full text of their review as indented text, interspersed with our responses addressing their specific comments as non-indented text and changes

to the manuscript in *italicised* font.

The paper "Intercomparison of TCCON data from two Fourier transform spectrometers at Lauder, New Zealand" by Pollard et al. presents an intercomparison of two high-resolution Fourier transform spectrometer measurements to assure the continuity of the Lauder TCCON data. Pollard et al. demonstrate that the difference between the column-averaged dry-air mole fraction of carbon dioxide (XCO2) data obtained from the two instruments is well below the uncertainty of the TCCON product.
The Lauder TCCON data have been widely used for carbon cycle studies and validation of satellite-based greenhouse gas and carbon monoxide measurements. The topic of this paper is significant for those research fields and well suited to Atmospheric Measurement Techniques. This paper is concisely written and contains a full description of the instrumental intercomparison. I therefore recommend publication of this paper after correcting and addressing several minor concerns below.

—

Specific comments
L80-81: Xair is scaled by the O2 column because Equation (2) can be rewritten as follows:

$$X_{air} = \left( VC_{air} - VC_{H_2O}\frac{m_{H_2O}}{m_{air}} \right) \frac{0.2095}{VC_{O_2}} \tag{1}$$

The reason Xair is used as a diagnostic of the measurement system is that the ratio between the retrieved columns is not taken for Xair.

Thank you for pointing out this error. We have replaced the sentence at L80-81 with:
*"The value and stability of $X_{air}$ is used as a diagnostic of the measurement system as $VC_{air}$ is independent of the instrument system and instrumental biases are not removed by scaling. Therefore deviations from the nominal value can be indicative of instrumental and systematic problems such as timing or pointing errors."*

L146: The median shift relative to the central wavenumber $\Delta\nu/\nu$ is $-0.469\times 10^6$ ($-0.469\,\Delta\nu/\nu \times 10^6$ is not the median shift). In addition, please define the variables $\Delta\nu$ and $\nu$ or $\Delta\nu/\nu$.

This sentence has been modified to read *"For a one-month period of the intercomparison we find that the median shift relative to the central wavenumber ($\Delta\nu/\nu$) for the ll instrument is $-0.469\times 10^{-6}$ (standard deviation $0.028\times 10^{-6}$) and for lr is $-0.507\times 10^{-6}$ ($0.026\times 10^{-6}$)."* and the axis label of Fig. 4 amended accordingly.

L154: Please clarify what the solar gas shift (SGS) means, in relation to just above sentence [GFIT accounts for . . .].

We have moved the definition of SGS to the preceding paragraph to make it clear what it refers to.

L184: It is unclear why "a small difference in the computed airmass for forward and reverse scans" induces the difference between the $X_{gas}$ data from the two instruments. Do the authors mean "a small error in the computed airmass (i.e., an error in zero path difference time)"?

This sentence has been modified to read as follows: *"This is likely caused by small differences in the time it takes both instruments to conduct a measurement, due to slightly different firmware versions or hardware, leading to small errors in the computed airmass which differ in magnitude for the forward and reverse scans."* to clarify the source of the spread in $X_{air}$ values at high solar zenith angles.

> L186-187: Please cite references for the values of the expected uncertainty of the retrieval scheme (0.25%) and the target site-to-site bias (0.2%). Provided that there are expected uncertainties of the retrieval scheme and target site-to-site bias for $X_{CH_4}$ and $X_{CO}$, I recommend specifying a similar evaluation here.

The paragraph has been re-written and the expected retrieval uncertainty altered to 0.2% to be consistent with the wider literature, a discussion of the $X_{CH_4}$ results included and citations added to Wunch (2010) and Wunch (2015). The footnote to Tab. 3 has also been amended so as not to describe the site-to-site bias estimate as a target.

> L197: October 2018 - October and November 2018 (to be consistent with Abstract and Introduction)

This change has been incorporated into the revised manuscript.

> Caption of Table 1: Transform - transform

This has been changed in the manuscript.

**References**

Wunch, D., Toon, G. C., Wennberg, P. O., Wofsy, S. C., Stephens, B. B., Fischer, M. L., Uchino, O., Abshire, J. B., Bernath, P., Biraud, S. C., Blavier, J. F. L., Boone, C., Bowman, K. P., Browell, E. V., Campos, T., Connor, B. J., Daube, B. C., Deutscher, N. M., Diao, M., Elkins, J. W., Gerbig, C., Gottlieb, E., Griffith, D. W. T., Hurst, D. F., Jiménez, R., Keppel-Aleks, G., Kort, E. A., Macatangay, R., Machida, T., Matsueda, H., Moore, F., Morino, I., Park, S., Robinson, J., Roehl, C. M., Sawa, Y., Sherlock, V., Sweeney, C., Tanaka, T., and Zondlo, M. A.: Calibration of the Total Carbon Column Observing Network using aircraft profile data, Atmos. Meas. Tech., 3, 1351-1362, 2010.
Wunch, D., Toon, G., Sherlock, V., Deutscher, N. M., Liu, C., Feist, D. G., and Wennberg, P. O.: The Total Carbon Column Observing Network's GGG2014 Data Version, doi: doi:10.14291/tccon.ggg2014.documentation.R0/1221662, 2015. 2015.

---

## Author Comment (AC2) · 12 Jan 2021

We thank the reviewer for taking the time to review our manuscript and for their constructive and thought provoking comments.

Below we have included the full text of their review as indented text, interspersed with our responses addressing their comments as non-indented text.
Review of Pollard et al "Intercomparison of TCCON data from two Fourier transform spectrometers at Lauder, New Zealand" for AMT.

The paper by Pollard et al describes the intercomparison of two collocated Bruker FTIR high resolution spectrometers. The instruments are operated within the Total Column Carbon Observing Network (TCCON. This network has well controlled analysed procedures (GFIT suite of software), as well as agreed upon instrumentation (Bruker 125HR), and measurement protocols. The NZ team is very experienced in both measurements and analysis procedures demanded by TCCON. They are actively involved in the TCCON network in terms of running their own site and contributing to the success of this network. On this basis this team is well placed to compare these instruments, one a new introduced FTIR, comparing the new one with an older established dataset. Their attention to detail is very good.

The text is well written, and as far as this referee can find, only one misplaced word (remove the first "of" in line 102). The authors establish that the measurement conditions are such that the comparison of the two datasets is relatively straightforward, that is, the conditions under which the data is collected is very similar in terms of instruments, collocation, and hence atmospheric conditions. They systematically consider the important nuances that have been carefully scrutinised and worked through over the years within the TCCON community, including Ghosts, airmass dependence, frequency shifts, signal to noise etc. The paper demonstrates that under normal conditions experienced at Lauder these two instruments perform at a remarkably consistent level, more than meeting various TCCON metrics.

The only suggestion here is a straightforward statistical one. Since the
main product that is compared, the means of the various retrieved Xgas, a simple t-test would give a solid quantitative basis to the conclusion that both instruments are measuring the same thing. This paper is recommended for publication in AMT.

The spurious "of" at line 102 has been removed.

The authors have spent some time considering the reviewer's suggestion of including a t-test. However, for the reasons set out below, we have decided not to.

The purpose of the manuscript, and it's main conclusion, is to show that the TCCON data record at Lauder can be considered continuous across the change of instrument. This has been achieved by demonstrating that the difference between  $X_{gas}$  retrievals of both instruments is smaller than the likely uncertainty in the retrieval process and site-to-site biases, and so will not have an adverse effect on users of the data.

There will, however, always be small differences between the instruments and hence a bias between their results. This, combined with the large sample of ten-minute averages (N=833), and the effect that has in reducing the standard error of the mean (SE), means that a t-test will inevitably conclude that there is a difference between the two sets of measurements. Indeed, conducting a paired t-test on the two sets of  $X_{CO_2}$  values yields t(832) = 18.2 and  $p

add significant further insight.
**Histogram of XCO2 difference (LL-LR)**

Fig. 1.